# Structure-Guided Identification of Critical Residues in the Vacuolar Cation/Proton Antiporter NHX1 from *Arabidopsis thaliana*

**DOI:** 10.3390/plants12152778

**Published:** 2023-07-26

**Authors:** Belén Rombolá-Caldentey, Imelda Mendoza, Francisco J. Quintero, José M. Pardo

**Affiliations:** Instituto de Bioquimica Vegetal y Fotosintesis, cicCartuja, Consejo Superior de Investigaciones Cientificas, Universidad de Sevilla, 41092 Sevilla, Spain; mbromcal@gmail.com (B.R.-C.); i.mendoza@csic.es (I.M.); fjquintero@ibvf.csic.es (F.J.Q.)

**Keywords:** potassium, ion transporter, vacuole, protein structure, Arabidopsis

## Abstract

Cation/Proton Antiporters (CPA) acting in all biological membranes regulate the volume and pH of cells and of intracellular organelles. A key issue with these proteins is their structure–function relationships since they present intrinsic regulatory features that rely on structural determinants, including pH sensitivity and the stoichiometry of ion exchange. Crystal structures are only available for prokaryotic CPA, whereas the eukaryotic ones have been modeled using the former as templates. Here, we present an updated and improved structural model of the tonoplast-localized K^+^, Na^+^/H^+^ antiporter NHX1 of Arabidopsis as a representative of the vacuolar NHX family that is essential for the accumulation of K^+^ into plant vacuoles. Conserved residues that were judged as functionally important were mutated, and the resulting protein variants were tested for activity in the yeast *Saccharomyces cerevisiae*. The results indicate that residue N184 in the ND-motif characteristic of CPA1 could be replaced by the DD-motif of CPA2 family members with minimal consequences for their activity. Attempts to alter the electroneutrality of AtNHX1 by different combinations of amino acid replacements at N184, R353 and R390 residues resulted in inactive or partly active proteins with a differential ability to control the vacuolar pH of the yeast.

## 1. Introduction

Cation/Proton Antiporters (CPAs) play a major role in regulating the volume and pH of cells and their intracellular organelles. These antiporters mediate the exchange of monovalent cations Na^+^ and K^+^ with one or two H^+^ across the membrane with different stoichiometries. CPAs are divided into two main groups, CPA1 and CPA2, whose members often differ in their ion selectivity and electrogenicity. A key issue with these proteins is their structure–function relationships, since many CPA proteins have intrinsic regulatory features that are intimately connected to structural determinants. High-resolution structures are only available for CPA proteins of prokaryotic origin, and they have served as a template to model the eukaryotic counterparts. All CPA proteins are thought to share a similar transmembrane topology at the active center, known as the Nha-fold [1], in which two short, partly unwound helices in transmembranes 4 and 11 create two partial positive (TM4c and TM11p) and two partial negative (TM4p and TM11c) dipoles that are facing each other [2,3].

Most CPA1 proteins contain a conserved asparagine–aspartate pair (ND motif) at their Nha-fold in TM5, whereas CPA2 members present with an aspartate–aspartate pair (DD motif). This dichotomy in the CPA1/CPA2 division has often been correlated with electrogenicity, so that the DD motif was thought to allow for the simultaneous translocation of two H^+^. However, this classical view has recently been challenged. Extensive phylogenetic comparison, together with structure-guided mutagenesis, has indicated that the CPA1/CPA2 division only partially correlates with electrogenicity, and that this property is not due to this DD motif, as previously thought [1]. A rationally designed triple mutant, only one of which showed a change from the DD motif to the ND motif, successfully converted the electrogenic EcNhaA to a electroneutral form [1]. The electrogenicity of the CPA2 family member NapA from *Thermus thermophilus* relied on the presence of a conserved Lys residue in TM10, which, in electroneutral CPA1 members, is exchanged for an Arg [4]. Further, HsNHA2 is an electroneutral protein, despite the DD motif in the active center and has an Arg residue (R432) in the homologous position to K305 of the electrogenic NapA from *Thermus thermophilus*. Mutation K305R changed NapA to electroneutrality, but the converse mutation R432K in HsNHA2 failed to trigger electrogenic transport [4]. The lack of consistent outcomes from structure–function studies provides the basis for the uncertainties regarding electrogenic transport in CPA exchangers.

To date, no crystallographic structure has been determined for any eukaryotic NHE/NHX protein of the CPA1 family. Different topological models have been described for the Arabidopsis K^+^,Na^+^/H^+^ exchanger NHX1, with contradictory results [5,6]. The main controversy in these reports is whether the C-terminal tail is oriented towards the cytosol or the vacuolar lumen.

Using prediction tools and in silico modeling, the topology and 3D models of several plant proteins of the CPA family have been generated based on the known atomic structures of prokaryotic Na^+^/H^+^ antiporters that are phylogenetically related. Structures have been modeled for the tree *Populus euphratica* PeNHX3, using the structure of EcNhaA from *E. coli* as a template [7], and for CHX17 using the TtNapA structure of *Thermus thermophilus* as the template [8]. Moreover, the Arabidopsis AtNHX6 structure was modeled using the MjNhaP of *Methanococcus jannaschii* as the template [9]. However, the confidence in these structures was not assessed by experimental approaches corroborating the proposed structures. In this study, a model structure for AtNHX1 was generated to better understand the structure–function relationships of the protein, and a mutagenic analysis was carried out with conserved residues presumed to be important to protein activity. Several conserved amino acids that are essential for the activity of AtNHX1 were identified at the active site and the Nha-fold. These residues are the T156 and D157 of the TD-motif in TM4, D185 of the ND-motif in TM5, and arginines R353 and R390 in TM10 and TM11, respectively. However, residue N184 at the ND motif, which is highly conserved in electroneutral antiporters of the CPA1 family is, surprisingly, not essential to the activity of AtNHX1. Last, in agreement with the proposed function in planta, we show that AtNHX1 provides a H^+^ shunt at the tonoplast that regulates vacuolar pH in yeast cells.

## 2. Results

### 2.1. Topological and Ternary Structure Models

The AtNHX1 protein sequence was used as the query in the SwissModel web-server dedicated to the homology modeling of protein structures [10,11]. Top-ranked templates with known structures were the archaea proteins PaNhaP of *Pyrococcus abyssi* (PDB 4cza.1) [12], MjNhaP1 of *Methanocaldococcus jannichii* (PDB 4czb.1) [13], and TtNapA of *Thermus thermophilus* (PDB 4bwz.1.A) [14]. All these archaeal proteins had better scores than the archetypical bacterial protein NhaA of *E. coli* (PDB Aau5.1.A) [15] (Appendix A). The sequence alignments mainly included the N-terminal hydrophobic domain of AtNHX1, with a coverage ranging from 58 to 71% of the protein length, while the C-terminal was not aligned in any case due to the lack of sequence similarities between the eukaryotic protein and the prokaryotic proteins.

The AtNHX1 topology modeled after the NhaP-type proteins comprised thirteen complete TM segments, whereas the model based on EcNhaA had only twelve. However, large segments of the N-terminal and C-terminal portions of the AtNHX1 sequence were not covered by the alignment with EcNhaA, and this impeded the modeling of the complete hydrophobic domain of the protein. The proposed topology of AtNHX1 (Figure 1) consisted of two well-defined domains: an intramembranous hydrophobic core from residues A28 to T434 corresponding to the pore domain involved in ion transport, and a cytosolic C-terminal tail from amino acids K435 to A538. In this model, both N- and C-termini were cytosolic. The N-terminal portion of the pore domain likely consists of two intramembranous (IM) semihelices (IM1.a and IM1.b), followed by eleven TM segments with an antiparallel topology, so that the odd TM segments (TM3, TM5, TM7, TM9 and TM11) are oriented from inside (cytosol) to outside (vacuole), while the even TM segments (TM2, TM4, TM6, TM8, TM10 and TM12) are oriented from the vacuolar lumen towards the cytosol. These TM segments are connected by short loops of varied lengths, with these being the shorter of 4 amino acids (between TM4 and TM5), and the longest of 18 (between IM1.b and TM2). The C-terminal tail is the most variable region among plant NHX proteins and may integrate the signals for protein regulation, such as protein phosphorylation and the binding of calmodulin-like proteins [6,16,17].

The 3D structure models of the complete pore domain of AtNHX1 were generated using the modeling data retrieved from the SwissModel server, and then visualized with the Pymol software (Figure 2). The two templates with the highest scores corresponded to two phylogenetically related archaea proteins PaNhaP and MjNhaP1 (Appendix A). The TtNapA protein, despite being an electrogenic exchanger, is structurally more similar to PaNhap and MjNhaP1 than to EcNhaA [3]. The experimentally determined structures of PaNhaP and MjNhaP1 were aligned with each other, as well as the modeled structure of AtNHX1 for each template (Figure 2C). The differences observed between structures were negligible. We selected PaNhaP for further analyses because, unlike MjNhaP1, this protein can bind Ti^+^ (besides Na^+^ and Li^+^), whose ion radius (1.5 Ȧ) is similar to that of K^+^ (1.44 Ȧ) and larger than that of Na^+^ (1.12 Ȧ). AtNHX1 has a low Na^+^-K^+^ discrimination ability [18,19] and the cation binding pocket must fit either substrate ion.

The structural assembly of the active center for the EcNhaA protein, the first CPA family member to be crystallized, is known as the Nha-fold [3]. In this non-canonical conformation of TM helices, the helical secondary structure of TM4 and TM11 of EcNhaA is interrupted by extended chains in the middle of the membrane, leaving two short helices in each TM flanking the centrally located extended chains [3,15]. The interrupted helices of TM4 and TM11, comprising the Nha-fold, cross each other at the extended chains in the proximities of the active site, which is located in the TM5 of EcNhaA (Figure 1; Appendix A). As a result, two short helices, oriented either toward the cytoplasm (c) or toward the periplasm (p), create two partial positive (TM4c and TM11p) and two partial negative (TM4p and TM11c) dipoles facing each other [2,3]. These charges are compensated by residual D133 in TM4 and K300 in TM10, respectively, creating a balanced electrostatic environment in the middle of the membrane at the ion-binding site. This fold has been confirmed for all the crystallized CPA superfamily members and incorporated into the modeled structures [7,20,21]. Figure 3 illustrates the modeled TM segments forming the Nha-fold of AtNHX1 using PaNhaP as template, whereas Appendix A shows the models based on templates MjNhaP1 and TtNapA.

To evaluate the quality of the structural models generated for AtNHX1, several in silico analyses were followed, including the distribution of positively charged residues and the pattern of evolutionary conservation (see Section 4). The results showed that the model structure of AtNHX1 is compatible with the conservation pattern, in which the protein hydrophobic core is highly conserved while the residues facing the lipid bilayer or located in extramembranous regions are variable (Appendix A). Expectedly, the variability of peripheral amino acids was higher at the vacuolar side of the membrane than the cytosolic side.

### 2.2. Conserved Residues in the Pore-Forming Domain

The alignment of the primary structure of representative proteins of the CPA superfamily allowed the identification of highly conserved amino acids in the active center comprising the Nha-fold (TM4, TM5 and TM11) and the adjacent TM10 (Figure 4). The asparagine at position 184 (N184) and the aspartic acid at 185 (D185) in TM5 comprise the active site of the protein. This ND-motif is highly conserved among electroneutral CPA1 family members, as described for MjNhaP1 (N160-D161), PaNhaP (N158-D159), HsNHE1 (N266-D267) and all six members of the Arabidopsis NHX family (Figure 4). In the electrogenic members of the CPA2 family, e.g., EcNhaA and TtNapA, the ND-motif in TM5 is exchanged for a DD-motif (D163-D164 and D200-D201, respectively) (Figure 4). These results are consistent with the placement of AtNHX1 in the CPA1 clade and the electroneutral exchange activity [19,22].

Two highly conserved, positively charged amino acids, R353 in TM10 and R390 in TM11, were detected in AtNHX1 (Figure 4). Other electroneutral proteins of the CPA1 family also present conserved basic residues at equivalent positions, e.g., R425 and R458 in HsNHE1, and R432 and K460 in HsNHA2. In MjNhaP1 and PaNhaP, the first arginine is conserved in TM11 (equivalent to TM10 of AtNHX1) as residues R320 and R337, respectively. The crystal structures have shown that this conserved arginine interacts with a glutamate in TM6 (E156 in MjNhaP1 and E154 in PaNhaP; equivalent to E180 in TM5 of AtNHX1) that is also conserved in the CPA1 antiporters. In the models obtained for AtNHX1 based on templates PaNhaP and MjNhaP1, R353 seems to interact with E180 (Figure 5). Moreover, E180 interacts with N184 in the three AtNHX1 models. The second arginine, R390 in TM11 of AtNHX1, is also conserved (Figure 4), but little is known about the function of this conserved residue. In HsNHA2, this residue is exchanged for a Lys.

The conserved threonine and aspartate residues in TM4 form a TD-motif in the unwound chain (Figure 1), which is also present in EcNhaA (T132-D133), and in the archea proteins MjNhaP1 and PaNhaP. In both cases, bacterial and archea proteins, this TD-motif has been described as taking part in ion coordination [12,23]. Moreover, T129 and T131 of PaNhaP and MjNhaP1 interact by their side chain with residues N158 and N160, respectively. These asparagines do not participate in the coordination of the substrate ion but control access to the ion-binding site [12,23]. In the mammalian HsNHE1 and HsNHA2 and in prokaryotic TtNapA, this motif is not conserved. No conserved TD-motif was described in the modeled structure of PeNHX3, and instead the presence of a non-charged Tyr in position 149 was proposed to be a substitute for the Thr residue in the TD-motif [7]. However, the alignment of the CPA proteins shown in Figure 4 indicates that the TD-motif is indeed conserved in PeNHX3. In AtNHX1 this motif is conserved as well (T156-D157) and, according to all the models obtained for AtNHX1, there seems to be an interaction of T156 with N184 (Figure 5), as described previously for the archaea NhaP proteins.

### 2.3. Conserved Residues with Functional Roles

To determine the relevance of the conserved amino acids in the activity of AtNHX1, site-directed mutations were introduced at each residue judged to be structurally important and the resulting protein variants were tested in the yeast *Saccharomyces cerevisiae* as reported previously [18]. The yeast strain AXT3K lacks the plasma membrane Na^+^ efflux proteins ENA1-4 and NHA1, which renders these cells sensitive to Na^+^ [24], and the prevacuolar Na^+^/H^+^ exchanger ScNHX1/VPS44 that also makes mutant cells sensitive to the cationic drug hygromycin B (HygB) owing to a defect in vesicle sorting and the hyperpolarized plasma membrane [25]. 

The wild-type AtNHX1 and the mutant alleles were cloned in the yeast expression vector pDR195 and transformed in the yeast strain AXT3K to test the functionality of the mutant proteins in both solid and liquid media. Plasmid pDR195 carries the strong and mostly constitutive *PMA1* gene promoter to drive the expression of the recombinant protein. The expression of wild-type AtNHX1 improved AXT3K growth in the presence of 50 µg/mL of HygB (Figure 6). Cells transformed with the mutant alleles of AtNHX1 showed different phenotypes. The mutation of the D185 residue to Leu or Asn (D185L/N) generated non-functional proteins, as expected. Other mutations of residues at the core of the active center of the protein that are part of the Nha-fold (D157N, R353L) generated equally inactive proteins (Figure 6). These results indicate that the conserved residues of the active core in AtNHX1 have an important functional and/or structural role in the protein, and they are essential for protein activity under the tested conditions. Notably, the mutation in which Asp substituted for N184 (N184D) in the active center partly suppressed the sensitivity to HygB (Figure 6). This unexpected result implies that the highly conserved residue N184 is important but not essential for activity.

Previous reports have shown that the antiporters HsNHA2 and EcNhaA lost their transportation activities when the DD-motif of CPA2 proteins was substituted for an ND-motif [14,21]. However, the mutation N184D to generate a DD-motif only decreased AtNHX1 protein activity, but was still able to overcome the HygB sensitivity (Figure 6). Thus, the integrity of the ND-motif is not essential to AtNHX1 activity, but it is important when aiming to reach optimal functionality. Uzdavinyz et al. (2017) [4] have demonstrated that the electrogenic properties of CPA2 family members is not due to this DD-motif, as previously thought, but to the presence of a conserved Lys in TM10, which, in CPA1 members, is exchanged for an Arg that promotes electroneutral activity (Figure 4). This position in AtNHX1 is occupied by R353, which is coherent with the electroneutral exchange of AtNHX1. Mutagenic studies of this conserved Arg in TM10 in different CPA1 family members have been performed but the results were inconclusive [7,21,26]. To study the relevance of the conserved R353 in TM10 of AtNHX1, and to explore whether it was possible to make AtNHX1 electrogenic, a set of allelic variants was generated at the R353 position, so that all possible conditions of ND/DD motifs and R/K residues were studied: DD-Motif + R353/R390; ND-motif + R353K; ND-motif + R390K; DD-motif + R353K; and DD-motif + R390K. In this allelic series, the R390K mutation was used as control to determine whether the electrogenicity (if changed) was dependent only on residual R353 or the active center motif ND/DD. 

Functional assays using the AXT3K yeast strain would not directly allow for the determination of whether AtNHX1 was transformed into an electrogenic protein, but could show whether the new arrangement of catalytic amino acids was compatible with protein activity in vivo. As shown in Figure 7A, any allele bearing an R-to-K mutation generated an inactive protein in solid media conditions, irrespective of the presence of an ND- or DD- motif in the active center. The only mutant that remained active was the N184D single mutant, although with slightly less activity than the wild-type AtNHX1 protein, as monitored by yeast growth. These results meant that the presence of conserved arginines R353 in TM10 and R390 in TM11 were indispensable to protein functionality. Moreover, although a charged amino acid in the active center in TM5 is necessary for this activity, the positively charged N184 is not essential and can be exchanged for a negatively charged amino acid such as an Asp (N184D mutation) with minimal effects.

In the CPA2 protein family, there is an interaction between a conserved Lys in TM10 and the Asp residue of TM5 in the active center. This Lys does not participate in ion binding; rather, a competition-based transport mechanism has been suggested. The exchange cycle would start with a periplasmic/outer side open conformation of the protein, in which D164 (in EcNhaA) or D157 (in TtNapA) is unprotonated, whereas D163 and D156, respectively, are engaged in a salt bridge with K300 or K305, which stabilizes this conformation. At low H^+^ concentrations, the binding of ions to the DD-motif breaks the salt bridge and a conformational transition releases the transported ions to the cytosol. Deprotonated D163 and D156 form a salt bridge with Lys at TM10, inhibiting the reorientation of the unloaded transporter. When Na^+^ binds from the cytoplasm, the salt bridge is broken and a conformational transition allows for Na^+^ release at the periplasmic/outer side of the membrane [2,15,27]. The same mechanism has been described for PaNhaP and MjNhaP1 [23,28]. However, in PaNhaP and MjNhaP1, the Arg replacing this Lys does not interact with the ND-motif but forms an ion bridge to the neighboring conserved glutamate in TM5, while the conserved Asn in TM5 interacts with the conserved Thr in TM4.

The main feature of this competition-based transport mechanism is that it is self-regulatory, ensuring that transport activity is switched off at extreme pH values to prevent excessive acidification or alkalinization of the cytoplasm [28]. This is in accordance with the notion that the K300 in EcNhaA and K305 in TtNapA seem to play more than a functional role in the protein, and could be part of the pH activation mechanism, as the pH dependence of both in K300R and K305R mutants was shifted to the alkaline side by one pH unit in comparison to the wild-type proteins [2,27,29].

Based on these considerations, we aimed to determine whether the non-functional mutants of AtNHX1 in medium YPD plus HygB had a different behavior under acidic and alkaline conditions. To that end, similar assays were performed in liquid YPD media buffered at a different pH, with and without the addition of HygB. Of note is that the inhibitory effect of HygB was much higher in the medium buffered at pH 7 (Figure 7). Regarding the growth of yeast cells expressing mutant proteins, the results were similar in all tests: all alleles that included a R-to-K mutation (either in R353K or R390K) failed to suppress the HygB-sensitive phenotype of AXT3K (Figure 7), regardless of the external pH. There were also no differences in their growth without the antibiotic. Interestingly, the only active allele was N184D, but this behaved differently in the presence of HygB at different pHs. The N184D mutation was shown to be detrimental to the protein activity at pH 5 and pH 7, as indicated by the reduced ability to suppress the sensitivity to HygB (Figure 7B). No difference between wild-type and N184D proteins in supporting yeast growth with HygB was found at pH 6. Growth was also tested in medium supplemented with Li^+^, a toxic analog of Na^+^. The N184D allele was able to sustain growth, like wild-type AtNHX1, while the R-to-K mutants could not suppress the ion sensitivity (Appendix A).

When combined, the results of yeast growth under various conditions suggest that N184 in the active center might contribute to pH sensing. This is in accordance to what has been previously described for EcNhaA and TtNapA (electrogenic proteins with a DD-motif), which are more active at a basic pH in comparison to MjNhaP1 and PaNhaP (with the ND-motif), which have been shown to be more active at acidic pHs [4,30].

### 2.4. Vacuolar pH and AtNHX1 Mutants

The activity of AtNHX1 and AtNHX2 in planta constitutes a leak pathway for vacuolar protons and, together, these two proteins help to control the vacuolar pH [31]. To assess the roles of AtNHX1 and its different variants in the regulation of vacuolar pH (pHv) of yeast, the ratiometric fluorescein-based pH-sensitive dye, 2′,7′-bis-(2-carboxyethyl)-5-(and-6)-carboxyfluorescein (BCECF) was used. BCECF localizes into the vacuole of yeast when introduced in its acetoxymethyl ester form (BCECF-AM). Previous studies have reported pHv values between 5.5 and 5.9 in wild-type *S. cerevisiae* strains [32] and references therein]. Under acidic pH stress, pHv in the wild-type was around 5.28 ± 0.14, and upon alkali stress, the pHv raised to 5.83 ± 0.13 [33,34]. Overexpression of the ScNHX1 protein resulted in the alkalinization of the vacuole, whereas mutation *nhx1* mutant produced acidification [35,36]. pHv measurements in our experiments were taken from yeast cells in the early logarithmic phase and media was buffered to 6.0 with arginine. Fluorescence ratios were transformed to pHv values, following the described protocol [37].

Firstly, yeast strains differing in the presence or absence of NHX exchangers were compared. Mutations of the *nhx1* gene produced the vacuolar acidification of 0.3 pH units relative to the wild-type strain, and the expression of AtNHX1 restored wild-type values (Figure 8). These results validated the experimental approach. The comparison of pHv of yeast cells expressing various AtNHX1 mutants showed a consistent tendency in independent experiments. Representative data from one experiment are shown in Figure 8. As expected, the dead-protein mutant D185L had an acidic pHv similar to the empty-vector control. On the contrary, the N184D mutant had a near wild-type pHv, demonstrating again that the N184D mutant protein behaves as a wild-type.

Surprisingly, conservative mutations in the conserved arginines R353K and R390K showed intermediate pHv values halfway between the wild-type and the null mutant D185L and empty vector controls, suggesting compromised exchange activity and reduced H^+^ leak. This result is somehow in contrast with the null-mutant phenotype shown by these mutants in the HygB and Li^+^ tolerance test (Figure 7 and Appendix A). However, combining the R353K and R390K mutation with the N184D mutation, which had no effect per se, produced acidic vacuoles similar to loss-of-function mutants.

## 3. Discussion

### 3.1. Generation of Topological and Tridimensional Models Based on Phylogenetic Relatedness

The NHX exchangers of plants are secondary transporters of the cation–proton antiporter (CPA) family [38]. CPA antiporters are conserved across all biological kingdoms and play essential roles in pH, ion homeostasis and volume control. CPA1 antiporters are electroneutral and exchange one cation (K^+^/Na^+^) with one H^+^. The CPA1 family, in which plant NHXs are included, contains proteins PaNhaP, MjNhaP1 and HsNHE1. CPA2 antiporters, including EcNhaA from *E. coli* and TtNapA from *Thermus thermophilus*, are electrogenic, exchanging one Na^+^ with two H^+^ [3,4,14]. This family also contains eukaryotic proteins, e.g., HsNHA2, and the plant CHX clade [38,39].

The first atomic structure of a cation–proton antiporter (CPA family) was the Na^+^/H^+^ exchanger of *E. coli* NhaA (EcNhaA). The structural map of EcNhaA resolved a characteristic folding of the active center, named the Nha-fold, consisting of two unwound transmembrane stretches that cross each other in the middle of the membrane near the ion-binding site by their unwound region, creating an X-shaped structure. Later, the structures of two CPA1 members, MjNhaP1 and PaNhaP, were also obtained. Although they contain some functional and structural differences to EcNhaA, the overall structure is well conserved, including the Nha-fold of the active center. The information gained from these structures allowed for the in silico modeling and helped to obtain a better understanding of the architecture of the more complex eukaryotic CPA proteins, such as plant proteins *Populus euphratica* PeNHX3 [7], and of *A. thaliana* AtCHX17 [8] and AtNHX6 [9]. Most models were based on the first available structure, that of EcNhaA, even if it was not the most adequate model, since EcNhaA is a CPA2 protein. In our work, three structures available at the SwissModel database were selected as templates because they best fitted the AtNHX1 protein sequence. These templates were *Methanococcus jannichii* NhaP1 (MjNhaP1), *Pyrococcus abysii* NhaP (PaNhaP), and *Thermus thermophilus* NapA (TtNapA). The alignment of the primary sequence of AtNHX1 with each of these three proteins individually and the alignment of all proteins together evidenced the high conservation of the core structure of the family. 

The 3D structures obtained for AtNHX1 with each of the model templates were highly similar, with the main differences being a more lax or tight structure of the protein, which, in turn, was reflected in the possible interactions that were predicted to take place among residues in the active site of the protein. However, the general structure was the same in all cases, and the predicted transmembrane segments based on individual alignments the overall alignment overlapped in all cases. This allowed for the generation of a topological model with 12 TM segments in the hydrophobic N-terminal part, or pore domain, and a long hydrophilic C-terminal tail, with both protein ends being cytosolic. Yamaguchi et al. (2003) [6] proposed a topology of AtNHX1 comprising nine transmembrane domains, three hydrophobic regions not spanning the tonoplast membrane, and the hydrophilic C-terminal domain in the vacuolar lumen. However, this topology is incoherent, with a protease protection assay concluding that the C-terminal tail of AtNHX1 was cytosolic [40], and proteomic studies showing the presence of phosphorylated residues within the C-terminal tail of the Arabidopsis NHX1/2 proteins [16]; no intravacuolar protein phosphorylation has been described to date. Based on a sequence and structural conservation approach, the model proposed herein yields a topology that is consistent with the common topology model proposed for the CPA superfamily: 12 TM antiparallel segments with a hydrophilic and cytosolic C-terminal tail. An important corollary of this topology is that the interaction of AtNHX1 with CML18 cannot take place in the vacuolar lumen as reported [6].

While this work was in progress, the modeled structure of AtNHX1, generated using the AlphaFold AI software, was released [41]. The AlphaFold structure was aligned with the models previously created using the Swiss model repository (Appendix A). The alignment between the models confirmed the presence of the Nha-fold in the active site of the AtNHX1 protein, with the side chains of the conserved amino acids being directed toward the intramembranous pocket in the center of the protein. The hydrophobic pore domain of the protein consists of TM segments arranged in a similar antiparallel fashion. However, the 3D structure generated by the AlphaFold software shows that the C- and N-terminal ends of the protein are located in different compartments of the cell. This is a direct consequence of the two complete TM segments (TM1 and TM2) modeled by the AlphaFold software. In our model, these segments are considered to be two intramembranous (IM) semi-helices. However, these differences should not affect their protein function or regulation. The final orientations of the other TM segments do not change compared to the topology predicted in this manuscript, and there are no previous reports of regulatory or activity domains in this region. The predicted 3D-structures aligned with a high level of accuracy, especially for those models generated with other CAP1 protein templates.

### 3.2. Identification of Conserved Residues with Functional Roles

Integrating the primary sequence comparisons and the topological and ternary models, several residues with probable structural and functional significance were detected. The graphical representation of the TM segments in which these residues are located made evident that AtNHX1 present with an Nha-fold at the active center, as previously demonstrated for crystalized proteins of the CPA superfamily. The active center N184-D185 motif of the CPA1 electroneutral proteins is located in TM5 of AtNHX1. In the vicinity of the ND motif, the expanded sectors of TM4 and TM11 cross each other, and in these sectors, the conserved T156-D157 (TD-motif) of TM4 and R390 of TM11 are located. Moreover, TM10 is present in the surrounding Nha-fold, and the conserved R353 in this TM interacts with conserved residues in TM4 and TM11 (Figure 3 and Appendix A).

The predominant view has been that the presence of the DD or ND motif in the active center of the protein determined the electrogenicity of the antiporter [3]. The additional negatively charged Asp residue in the DD motif of CPA2 proteins was proposed to be responsible for the extra H^+^ that is transported by these exchangers compared with CPA1 members carrying the ND motif [3,30]. For both EcNhaA and TtNapA the DD motif is essential to the protein activity, and introducing an Asn residue to this position rendered the protein inactive [14,42]. In CPA1 protein members, the substitution of the Asp residue in the ND motif showed that it is essential for the protein activity. However, the Asn residue was not. Replacing N160 in MjNhaP1 for an Ala generates an inactive protein, but the N160D substitution mutant of this protein [26], as well as the N187D substitution in PeNHX3, generated active proteins [7]. Similar results were obtained for AtNHX1 since the N184D mutation also produced a biologically active protein, whereas mutations D185L and D185N produced inactive proteins (Figure 6). Nonetheless, Asn-to-Asp mutations in the active enter of CPA1 proteins do not render these proteins electrogenic [7,23,26]. In the case of AtNHX1, more analyses are required to confirm this. The N184D mutant was shown to be detrimental to yeast growth in the presence of hygromycin, which is indicative of compromised activity, but supported a slightly more robust growth than the wild-type AtNHX1 when grown in 10 mM LiCl (Figure 7 and Appendix A). These results indicate that the highly conserved N184 in the active center is not essential to the activity of AtNHX1, as previously demonstrated for MjNhaP1 or PeNHX3, but could have a role in substrate selectivity by changing the geometry of the active center. In MjNhaP1, it was proposed that the homologous Asn is necessary to stabilize the proton or substrate-bound state, which can also be fulfilled by Asp [23]. As for HsNHA2, the mutation of thee DD-motif into an ED-motif decreased Li^+^ tolerance, which is the opposite to what we observed when the ND-motif of AtNHX1 was exchanged for a DD-motif (Appendix A), implying again that the presence of an Asp in the first position of the DD motif can modify ion affinity and/or selectivity [21].

Residue R353 in AtNHX1 located in TM10 has counterpart homologues in MjNhaP1 (R320) and PaNhaP (R337), both in TM10. The X-ray structure of these proteins shed light on their function in these proteins. These Arg residues form ion bridges with a conserved Glu in TM5: R320 with E156 in MjNhaP1, and R337 with E154 in PaNhaP. This Arg-coordinating Glu is conserved in all CPA1 antiporters, and the corresponding residue in AtNHX1 is E180. These highly conserved residues, which are essential to the activity, seem to have a role in stabilizing the protein [13,26]. Recent studies showed that the electrogenic activity of TtNapA is due to the presence of a Lys in TM10 (K305) but not to the DD-motif in the active center [4]. This Lys is the equivalent to residue R353 of AtNHX1. They also demonstrated that HsNHA2 is an electroneutral protein despite the DD-motif in the active center and because of the presence of an Arg (R432) in the homologous position to K305 of TtNapA. The equivalent Lys in EcNhaA (K300) is not essential to electrogenic transport [2]. Together, these results indicate that the DD motif is not the basis for electrogenicity and that the K/R dichotomy in TM10 only partially explains this transport mode.

In the CPA2 family of proteins, there is an interaction between the conserved Lys in TM10 and the Asp residue in the active center, and this interaction regulates the pH sensing and the activity of the protein in a competition-based transport mechanism that ensures transport activity as long as no extreme pH values are reached, in order to prevent excessive acidification or alkalinization of the cytoplasm [2,15,27,28]. In the case of PaNhaP and MjNhaP1, the Arg replacing the Lys does not interact with the ND-motif but forms an ion bridge to the neighboring conserved glutamate in TM6, and the conserved Asn of the ND-motif in TM5 interacts with a conserved Thr in TM4. The essential role of R353 in protein function or stability has been confirmed by functional analysis in yeasts (Figure 6 and Figure 7). In the mammalian CPA1 proteins NHE1 and NHA2, the Arg in TM10 is conserved (Figure 4) but, unlike the AtNHX1 mutant R353K, the R-to-K mutant of NHA2 retained partial activity. Something similar occurs with the conserved positively charged residue in TM11 (R458 in HsNHE1; K460 in HsNHA2; R390 in AtNHX1). The mutant K460A of HsNHA2 exhibited a Li^+^-selective phenotype, signifying that the mutant protein had greater affinity for Li^+^ than the wild-type protein [20,21]. In prokaryotes, the functionally important Arg in the unwound TM11 (R285 in MjNhaP1; R362 in PaNhaP; R390 in AtNHX1) might also have a stabilizing function by interacting with residues in the active site [12,23], but not much is known about them. Our data show that even the conservative change R390K was unable to recover the yeast sensitivity to HygB (Figure 7), indicating that R390 might also have a functional or structural role in the AtNHX1 protein.

In EcNhaA, the conserved residues T132 and D133 (TD-motif) are involved in the ion coordination and translocation, but they are not essential for protein activity [27,43]. Instead, a function in the stabilization of the active site was proposed [27] and recently confirmed [44]. In the archea proteins PaNhaP and MjNhaP1, the TD motif is conserved and is essential to the coordination of the ion [12,23]. In PaNhaP and MjNhaP1 residues, Thr129 and Thr131 have been described to interact through their side chain with N158 and N160, respectively, which do not participate in the ion coordination but control the access to the ion-binding site. In the mammalian NHE1 and NHA2, and in TtNapA, this TD-motif is not conserved. Although, for PeNHX3, only the presence of a non-charged Tyr in position 149 was described [7], the alignment of the CPA proteins performed herein indicates that the TD motif is, in fact, conserved (Figure 4). This motif is also conserved in AtNHX1 (T156-D157) and, according to all the models obtained for AtNHX1, there seems to be an interaction of T156 with N184, as described for the archea NhaP proteins. Our yeast complementation assays showed that D157 of the TD motif is essential to AtNHX1 activity, but no mutant in T156 was tested.

The AtNHX1 topological and tridimensional structure generated by homology modeling in this study sheds light on its transport mechanism. In addition, the analysis of the most conserved residues demonstrates that AtNHX1 maintains essential properties of the CPA1 family, but also carries unique features in ion binding and translocation. Additional ion transport assays in tonoplast vesicles should be carried out to confirm the functionality of the described amino acids in the electroneutral nature of AtNHX1 or in the affinity traits.

A relevant issue is the structural basis for substrate selectivity. Endosomal proteins AtNHX1-6 are non-selective exchangers of monovalent alkali cations (K^+^, Na^+^, Li^+^) [19,45,46], whereas AtNHX7/SOS1 is selective for Na^+^ and Li^+^. AtNHX8 has been suggested to be a selective Li^+^ transporter [47,48,49]. Ion selectivity may be affected by interacting partners [50] and modified by mutagenesis [18,51]. The current model proposes that amino acids surrounding the TD-motif in TM4, together with the ND-motif at the catalytic center of TM5, determine the substrate specificity of cation/proton exchangers of the CPA superfamily [1]. The Na^+^-selective PaNhaP and MjNhaP1 antiporters contain the ATDP sequence in the TD-motif of TM4, in which the Pro residue plays a structural role in the ion coordination pocket, whereas the STDA motif is an important determinant in the selective coordination of K^+^ ions, providing an additional bond with the transported ion. The ATDP motif is indeed present in the Na^+^-selective proteins SOS1/NHX7 and NHX8. However, the corresponding sequence of the vacuolar non-selective N,K/H antiporters AtNHX1 and AtNHX2 is ATDS. In would be interesting to conduct a domain-shuffling to test whether discrete changes in the TD- and ND-motifs are sufficient to determine substrate specificity.

### 3.3. Regulation of Vacuolar pH by AtNHX1

One of the main functions of plant NHX proteins is to regulate the luminal pH of organelles in which they reside [31,52,53]. This function is highly conserved in eukaryotic organisms, including yeasts in which the *nhx1* mutant has been shown to have a more acidic endosomal pH when subjected to acid stress [35,36]. The yeast ScNHX1 protein resides in late endosomes/pre-vacuolar compartments, and ScNHX1 effects on organelle pH seem to be tied to intracellular trafficking. The *nhx1* mutant is also known as vacuolar protein-sorting 44 (*vps44*) [25]. The *vps44/nhx1* mutant secreted 35% of a vacuolar carboxypeptidase Y (CPY) and missorted markers associated with PVC, Golgi, or the vacuolar membrane. These results show that (Na^+^,K^+^)/H^+^ exchange activity is essential to endosomal function and protein-sorting in eukaryotic cells, and that proper pH homeostasis in LE/PVC is critical for the sorting of proteins.

When extracellular pH approaches the cytosolic pH, nutrient and ion uptake can be disrupted because the pH gradient across the plasma membrane is lost. The *ENA1* gene, encoding a Na^+^-ATPase, is induced by alkaline conditions and encodes a pump capable of exporting toxic Na^+^ in the absence of an H^+^ gradient [54]. This protein is mutated in strain AXT3K, which may explain why reduced yeast growth was observed at pH 7.0 (Figure 7). The expression of AtNHX1 conferred yeast tolerance to high K^+^ or Na^+^ at an acidic external pH, but on alkaline medium, AtNHX1 was ineffective [39]. Attempts to monitor AtNHX1 in yeast, based on resistance to hygromycin instead of to Na^+^ or Li^+^, were also unsuccessful because, for unknown reasons, the toxicity of hygromycin increased greatly at pH 7 (Figure 7).

Measurements of luminal vacuolar pH (pHv) in yeast cells by the pH-sensitive fluorochrome BCECF showed that vacuoles of the *nhx1* mutant were more acidic than those of the wild type (Figure 8), which is consistent with the idea that ScNHX1 catalyzes K^+^ uptake and H^+^ efflux from compartments acidified by the vacuolar H^+^-pumping ATPase [35]. Although quantitative estimations of pHv varied from experiment to experiment, trends in pHv differences between genotypes were maintained. Important differences were observed in the pHv of strain AXT3K (*nhx1*) expressing various AtNHX1 mutants affected by the conserved residues. The vacuole of yeast carrying the AtNHX1 mutant N184D had a more basic pHv than AXT3K, similar to the wild-type AtNHX1, which correlates with the ability of this mutant to suppress sensitivity to HygB. In CPA1 proteins, the conserved Asn in the active center (ND-motif) may interact with the TD-motif at the Nha-fold (Figure 3 and Appendix A). The exchange of Asn by Asp in the N184D mutant of AtNHX1 might alter this interaction and maintain the protein’s active state at the optimal external pH 6, but it seems to be detrimental to growth at pH 5 (Figure 7). The equivalent N187D mutant of PeNHX3 also complemented the yeast *nhx1* mutant, but the vacuolar pH was not measured [7]. In MjNhaP1, the N160D mutant showed lower activity compared to the wild-type [23]. 

Surprisingly, single mutants with the two conserved arginines R353 and R390 changed to Lys showed an intermediate pHv compared to the wild-type AtNHX1 and a loss-of-function mutant (Figure 8). This was unexpected because, in the growth assays, no complementation of the AXT3K sensitivity to HygB or LiCl by R353K and R390K mutants was detected. The double mutants, in which the ND-R arrangement of AtNHX1 was completely changed to the DD-K configuration found in electrogenic CPA2 proteins, showed a pHv similar to loss-of-function mutants, e.g., D185L (Figure 8). The phenotype of yeast expressing the R-to-K variants could be explained by the lower activity of the AtNHX1 protein being insufficient to suppress the phenotype of the yeast *nhx1* mutant and yet enough to constitute a H^+^ shunt, preventing extreme acidification of the vacuolar lumen. Although, under normal growth conditions, these proteins are able to translocate K^+^ into the vacuole in exchange of H^+^, the selective conditions applied in the functional assays in yeast could still inhibit the yeast growth, thereby masking the reduced protein activity. This low activity of R353K and R390K mutants could be explained by a change in ion selectivity or the inability to properly coordinate the substrates due to an altered, less than optimal structure. This could also explain the sensitivity of AXT3K cells expressing R353K and R390K mutant proteins in AP media with LiCl. Moreover, in the event that the nature of the mutant AtNHX1 protein changes from electroneutral to electrogenic, it could be deleterious to yeast in selective conditions even though the protein remains active. Nevertheless, all these suppositions are only based on indirect evidence gathered from growth assays and measurements of the vacuolar pH as affected by the AtNHX1 mutants. A direct assay measuring the ion transport of the mutants in tonoplast vesicles is necessary.

In TtNapA, the mutation of the conserved K305 for other basic amino acids, such as Gln or Arg, generated alleles that were only slightly active at pH 8. Moreover, the new alleles lost their electrogenic properties and their activity became electroneutral. The activity could only be recovered to wild-type levels in a D156N-K305Q mutant. This made evident the need for interaction between a positive amino acid that differs from Lys in TM10 and the Asn in the active center to ensure the activity of electroneutral proteins (CPA1). This double mutant was also electroneutral, and the mutation from K to R alone was enough for this conversion. Recently, it has been reported that EcNhaA could be converted from electrogenic to electroneutral by mutating the D163 in the DD motif into Asn (ND motif), together with two other amino acids that differ to the conserved amino acids mentioned here, but with strategic positions near the active center (A106S and P108E) [1]. These results demonstrate that not only is the presence of certain conserved amino acids essential to the activity of the protein, it is also essential to maintaining their interactions to maintain the necessary conditions in the active site for the kind of transport that takes place. However, according to Uzdavinys et al. (2017), [4] the conserved Lys in TM10 is essential for the electrogenicity of the protein, while for Masrati et al. (2018) [1], it is the first Asp of the DD motif. Nonetheless, in both cases, it was possible to mutate the active site of an electrogenic DD motif into a ND motif, obtaining an active protein that was always balanced by mutations of other strategic residues that maintained the correct conditions for the translocation of ions, generating electroneutral proteins from a electrogenic one. This would mean that converting AtNHX1 into an electrogenic state would probably require more than one mutation. Based on published results with other CPA proteins, the N184D-R353K mutant of AtNHX1 might act as electrogenic, but transport assays in yeast vesicle should be carried out to demonstrate this point. 

In conclusion, the topological and ternary models of AtNHX1 indicate that this plant protein conserves the structural features characteristic of microbial and mammalian members of the CPA superfamily. Several conserved amino acids that are essential for the activity of AtNHX1 have been identified at the active site and the Nha-fold. These residues are D157 of the TD-motif in TM4, D185 at the ND-motif in TM5, and arginines R353 and R390 in in TM10 and TM11, respectively. Residue N184 at the ND-motif, which is highly conserved in electroneutral antiporters of the CPA1 family, is not essential to the activity of AtNHX1, at least in the heterologous system used to validate the functionality of mutated AtNHX1 proteins. Last, in agreement with the proposed function in planta, AtNHX1 can regulate vacuolar pH in yeasts cells, and the expression of proteins mutated in the conserved amino acids of the Nha-fold generates pH differences in the yeast vacuole.

## 4. Materials and Methods

### 4.1. Structural Models for AtNHX1 and In Silico Validation

A BLAST search was conducted against the SwissModel database [11] using the AtNHX1 protein sequence as query to find templates and generate structural models. All pairwise alignments and multiple alignments of proteins sequences created in this project were established using the default parameters of the MUSCLE software [55]. Visual 3D model structures were generated using the PyMol software v1.8. The HMM was generated using the HMMblits tool from the Bioinformatics department of the Max Planck Institute for Developmental Biology, Tübingen (https://toolkit.tuebingen.mpg.de/#/tools/hhblits, accessed on 15 October 2018).

To evaluate the quality of the structural models generated for AtNHX1, several in silico analyses were carried out. Analysis of charge distribution in the AtNHX1 three-dimensional structure modeled with templates TtNapA, MjNhaP1 and PaNhaP followed the rule that the intracellular positions of intrinsic membrane proteins are enriched with positively charged residues (Arg and Lys) compared to extracellular regions [56] (Appendix A). The polar residues were clustered either in the inner structures of the protein or at the extramembranous loops, maximizing the exposure of hydrophobic residues to the membrane lipids (Appendix A). The conservation of residues subjected to evolutionary pressure was calculated using the ConSurf server (http://consurf.tau.ac.il/, accessed on 15 October 2018) [57]. To generate a conservation model, three different approaches with increasing levels of stringency were used. In brief, the AtNHX1 sequence was used as a query in the UNIPROT database [58], using PSI-BLAST [59] to collect homologous sequences. Redundant results (>95% sequence identity), or those with low coverage of the protein (<60% identity), as well as fragmented sequences, were discarded. The resulting 216 sequences were aligned using MUSCLE [55] with default parameters, and the final Multi-Sequence Alignment (MSA) [59] was used to generate a Hidden Markov Model [60]. The generated sequences were subsequently used to generate a new alignment in the MUSCLE server under default conditions (Appendix A), or to collect remote homologous sequences from the UNIPROT database using PSI-BLAST, aligned using MUSCLE (Appendix A). The final alignment including proteins from all kingdoms was exclusive to Na^+^/H^+^ exchangers related to AtNHX1, and highly reliable to infer position-specific evolutionary information for this transporter. A final step was carried out using only the AtNHX1 PDB sequence generated by SwissModel and the ConsurfServer default conditions to generate the MSA for the conservation analysis (Appendix A). Based on the MSA obtained in steps *a* and *b* (formed by 352 and 243 sequences, respectively), or the MSA defined by the ConsurfServer-sequence, evolutionary conservation scores were calculated using a Bayesian method [61] and the ConSurf web-server. The obtained scores were projected onto the 3D model of AtNHX1.

### 4.2. Plasmid Constructs

The mutant alleles of *AtNHX1* generated in this study are listed in Appendix A, together with the primers used to produce the point mutations by PCR amplification. For radical changes in amino acidic residues, polar residues were changed to Asn or Leu (D157N, D185L/N, R353L) based on the dissimilarity index D [62]. Mutation N184D was made to create the DD motif found in most electrogenic CPA2-type exchangers. Conservative mutations R353K and R390K were made to test whether the strict conservation of arginine residues was indispensable for activity or whether they could be replaced by another basic residue. To generate mutated alleles, two different strategies were followed. In all cases, the pBluescript (KS)-NHX1 plasmid was used as template. For the first set of mutants, the New England Biolabs’ Q5^®^ Site-Directed Mutagenesis Kit was used, following the manufacturer’s specifications. Primers for this protocol were designed using the online tool supplied by NEB (http://nebasechanger.neb.com, accessed on 15 October 2018). The second set of mutants was obtained using a high-fidelity polymerase according to [63]. Primers for these reactions were designed so that they had the desired mutation, flanked by the 10 bp template fitted in both directions; the reverse primer was the reverse complementary sequence of the forward primer. As both primers fit each other better than to the template, two reactions were set, each with only one of the primers, for five cycles. Afterwards, 10 μL of each reaction was mixed and, after addition of the polymerase, the combined reaction was allowed to proceed for 13 cycles. The elongation time was chosen to allow for the whole template plasmid to be amplified. After the PCR reaction, 10 μL of the product was treated with 0.5 μL of *Dpn*I, which cuts only the methylated GATC sites, in the PCR buffer for 1 h 37 °C. *E. coli* cells were transformed with 5 μL of the digestion. The generated alleles were subcloned in plasmid pDR195 as *Xho*I/*Not*I fragments and confirmed by sequencing. Yeast transformations were performed using the lithium acetate/PEG method [64].

### 4.3. Complementation and Growth Assays

Yeast growth assays were used to determine the tolerance of AXT3K yeast transformants to NaCl or LiCl salts in Arginine–Phosphate (AP) medium (8 mM H_3_PO_4_, 10 mM L-arginine, 2 mM MgSO_4_, 0.2 mM CaCl_2_, 1 mM KCl, 2% glucose, 1% oligoelements, 1% vitamins, pH 8.5 with L-arginine) [65], and to hygromycin B in YPD medium. Yeast was transformed with the NHX1 alleles cloned in plasmid pDR195 and selected in YNB plates without histidine and uracil. Yeast transformants were inoculated in 2 mL of YNB media, supplemented with the corresponding amino acids and grown at 30 °C overnight. Cells were then harvested and resuspended to a final OD_600_ of 0.5. The cells were then serially diluted 10-fold in water, and 5 μL of each dilution was spotted onto the selective media. Plates were incubated at 30 °C for 2–3 days. 

For growth assays in liquid media, the transformed yeast cells expressing the different NHX1 alleles in plasmid pDR195 were inoculated in 2 mL of YNB media, supplemented with the corresponding amino acids and grown at 30 °C overnight. The cells were then harvested and resuspended to a final OD_600_ of 0.5. Next, 20 μL of this dilution was used to inoculate a 96-well plate containing 200 μL of liquid media per well. From the first well, 10-fold serial dilutions were carried out. Plates were incubated at 30 °C for 2–3 days. The OD_600_ of the culture was measured 24 and 48 h after inoculation with the Varioskan LUX Multimode Microplate Reader (ThermoFisher Scientific; Waltham, MA, USA). Three independent AXT3K transformant colonies were used in each assay.

### 4.4. Measurements of Vacuolar pH

Measurements of the vacuolar pH were carried out in a microplate reader. Yeasts were grown overnight in AP pH 6.0 medium without histidine and uracil. Cultures in the exponential phase of growth (OD_600_ 0.5–0.6) were harvested, washed twice and finally resuspended in AP medium without amino acids to a final OD_600_ of 0.2–0.3. Three independent colonies per mutant allele and controls were used. Each sample was incubated with 50 µm of 2′,7′-Bis-(2-carboxyethyl)-5-(6)-carboxyfluorescein acetoxymethyl ester (BCECF-AM; Molecular Probes) at 28 °C, with gentle shaking. After 20 min, the culture was centrifuged 10′ at 5000 rpm and washed three times with AP medium without amino acids and without BCECF-AM (incubating during 10 min with the new medium after each centrifugation). Finally, yeast cells were resuspended in 100 μL of AP medium without amino acids and without BCECF, and the fluorescence was measured. Fluorescence intensity and absorbance values were recorded using the Varioskan Microplate Reader at 20 °C that was shacked before each measurement. Samples were sequentially excited by two wavelengths, 450 nm and 490 nm; emission fluorescence was detected at 535 nm for each of the two excitation wavelengths. Absorbance was also measured at 600 nm. Three reads per well and wavelength were taken. Measurements were repeated three times for each culture, washing the cells in between with AP medium without amino acids and without BCECF. At the end of each experiment, a calibration curve of fluorescence intensity versus pH was obtained. Samples were centrifuged and resuspended with 200 μL of the following calibration medium: 50 mM MES, 50 mM HEPES, 50 mM KCl, 50 mM NaCl, 200 mM ammonium acetate, 10 mM NaN_3_, 10 mM 2-deoxyglucose, 50 µM cyanide m-chlorophenylhydrazone. Buffers were titrated to eight different pH values (5.2, 5.6, 6.0, 6.4, 6.8, 7.2, 7.6, 8.0) using 1 M NaOH. Settings of the Microplate Reader were identical for measurements and pH calibration.

To accurately estimate acid pH values below 5.0, the approach of James-Kracke (1992) [37] was followed. According to this, at a pH close to neutrality, BCECF fluorescence measured as H^+^ activity or H^+^ concentrations using the following formula:(1)$$H^{+}=K_{a}x\frac{\left(R_{max}-R\right)}{\left(R-R_{min}\right)}x\frac{F_{base450}}{F_{acid450}}$$
with *K_a_* being the acid dissociation constant, *R* the ratio of the emitted fluorescence by BCECF excited at 490 and 450 (Ratio 490/450), *R_max_* the maximum 490/450 ratio value, obtained in alkaline conditions, and *R_min_* the minimum 490/450 ratio in acid conditions. *F_base_*_450_/*F_acid_*_450_ is the ratio of the fluorescence at 450 nm in acid and basic conditions.

The logarithmic transformation of the equation is: (2)$$pH=pK_{a}-log\left(\frac{F_{base450}}{F_{acid450}}\right)-\log\left(\frac{\left(R_{max}-R\right)}{\left(R-R_{min}\right)}\right)$$

In the isosbestic point *F_base_*_450_*/F_acid_*_450_ is 1, this can be explained by the following equation:(3)$$pH=pK_{a}-\log\left(\frac{\left(R_{max}-R\right)}{\left(R-R_{min}\right)}\right)$$
where *R_max_* and *R_min_* are obtained from the most alkaline and most acidic pH buffers that were measured. 

The values obtained for each culture treated with different buffers were background-subtracted and normalized to cell density. The pH calibration curve was obtained by plotting $\log\left(\frac{\left(R_{max}-R\right)}{\left(R-R_{min}\right)}\right)$ against pH. The resulting equation was used to obtain the *pKa*. After blank subtraction (yeast without BCECF treatment) and cell density normalization, to obtain pH values, the ratio 490/450 was calculated in each case, and the obtained values were extrapolated from the calibration curve. The data were analyzed with the Microsoft Excel software. Statistical analysis was performed using the OriginPro 2017 program.

## 5. Conclusions

Previously modeled structures of NHX-type vacuolar exchangers, belonging to the CPA1 family, were based on the crystal structure of EcNhaA, an electrogenic CPA2 protein. Here, we produced a high-confidence structural model of the Arabidopsis NHX1 family based on the structures that were experimentally determined for microbial CPA1 proteins. Site-directed mutagenesis and mutant complementation analyses suggested that the residue N184 in the canonical ND motif of CPA1 proteins is not essential, and that the ND motif can be converted to the DD motif of CPA2 proteins without major consequences for protein activity. Based on evolutionary and structural considerations, we can conclude that residues R353 and R390 are critical to the activity of AtNHX1. Finally, we can show the plant transporter AtNHX1 is able to modulate the vacuolar pH of yeast cells, thereby providing a convenient platform to analyze the transport activity of NHX-type transporters in vivo.

## Figures and Tables

**Figure 1 plants-12-02778-f001:**
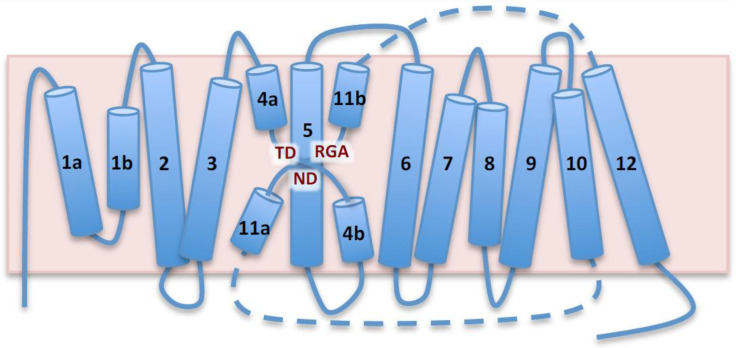
Topological model of AtNHX1 based on prokaryotic templates. Schematic representation of AtNHX1 topology based on structural similarities to PaNhaP, MjPhaP1 and TtNapA templates. AtNHX1 consists of two domains, a membranous hydrophobic pore domain, shown in the diagram, and a C-terminus hydrophilic tail (not shown). Both the N- and C-termini are cytosolic. Barrels represent transmembrane (TM) segments numbered 1a,b to 12, and the pink box is the tonoplast. The pore domain is formed by two intramembrane helices, TM1a and TM1b, followed by eleven TM segments with antiparallel orientation. The semi-helices TM4a,b and TM11a,b, crossed over TM5, constitute the catalytic core known as NhaA-fold. Dashed lines represent the loops connecting TM10, TM11a,b and TM12, and they are not drawn to scale because TM11a,b is placed in the vicinity of TM4a,b and TM5 to illustrate the NhaA-fold. Conserved amino acids in the NhaA-fold are highlighted.

**Figure 2 plants-12-02778-f002:**
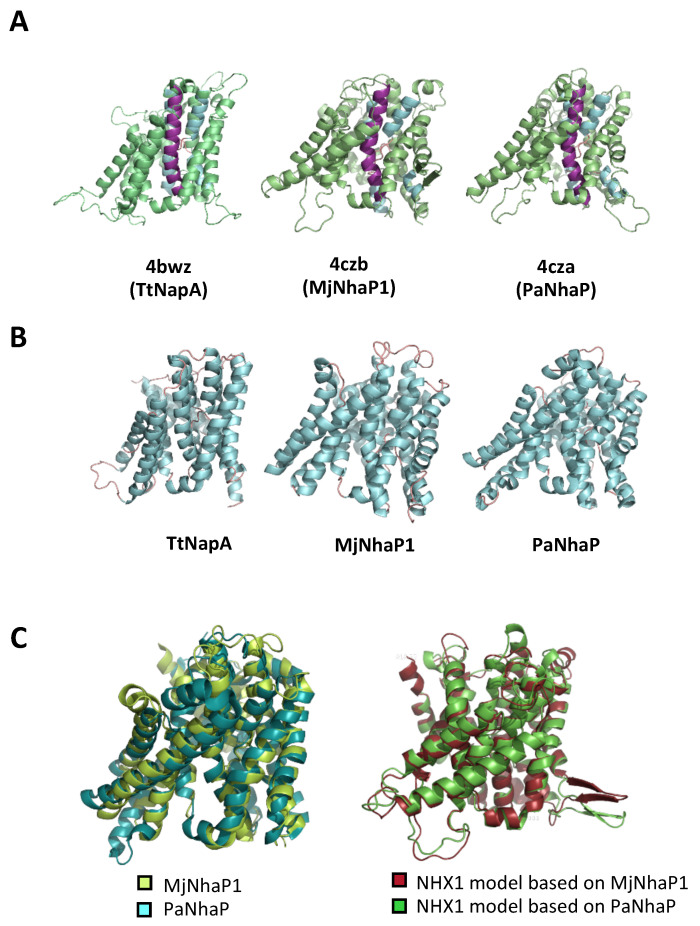
Models of AtNHX1 structure based on crystalized prokaryotic templates. (**A**) Tridimensional structural representation of AtNHX1 models produced by SwissModel using the indicated structural templates. TM5 is highlighted in purple; TM4 and TM11, in blue. (**B**) Crystal structure of the prokaryotic template proteins. (**C**) Structural overlay of the archea proteins MjNhaP1 and PaNhaP (**left**), and of the AtNHX1 models based on these template structures.

**Figure 3 plants-12-02778-f003:**
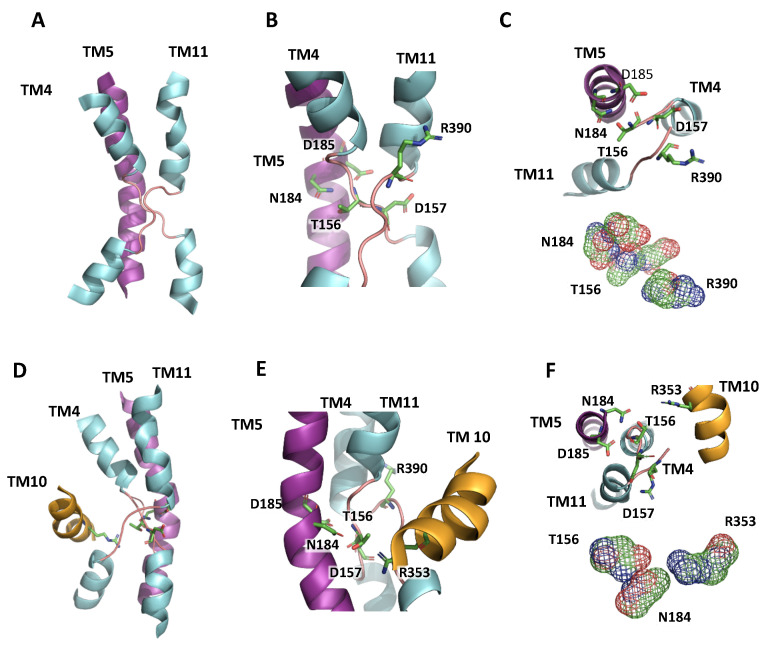
NhaA-fold in the active center of AtNHX1. (**A**) Representation of TM4, 5 and 11 in the AtNHX1 structure, showing the NhaA-fold in the active center of the protein, modeled with PaNhaP as a template. (**B**) Close-up view of the cross-over of the extended chains in TM4 and TM11, indicating the highly conserved residues T156 and D157 in TM4, N184 and D185 in TM5, and R390 in TM11. (**C**) A top-view (inside to outside) to show how D157 and R390 side chains compensate the dipoles generated by the crossing of TM4 and TM11. Nearby, T156 and N184 side chains are close enough to interact. (**D**) Representation of the active center with the NhaA-fold and TM10. (**E**) Closer view of the NhaA-fold and the localization of R353 in TM10 relative to the active center. (**F**) A top-view (inside to outside) to show the interaction of N184, T156 and R353 side chains.

**Figure 4 plants-12-02778-f004:**
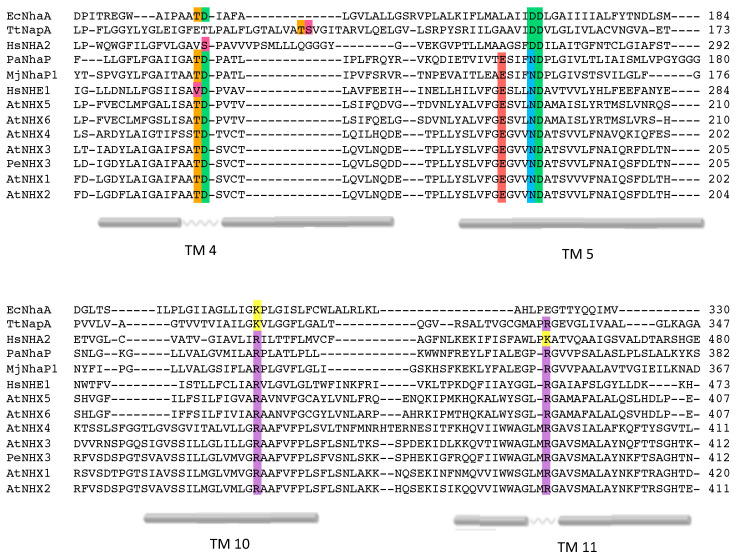
Alignment of AtNHX1 sequence with prokaryotic templates. Sequence alignment of CPA1 and CPA2 proteins cited here. The most relevant conserved amino acids and the TM segments in which they are located are highlighted with different colors according to their side chain group. TM numeration is in accordance with the topological model of AtNHX1 shown in Figure 1.

**Figure 5 plants-12-02778-f005:**
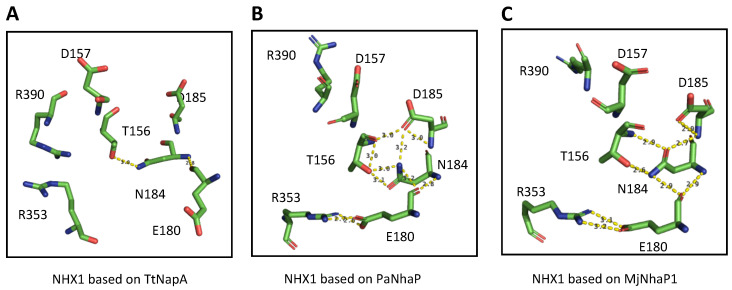
Predicted amino acid interactions in the active site. Representation of the side chains of the amino acids proposed to affect the activity of AtNHX1 according to the model obtained from the (**A**) TtNapA, (**B**) PaNhaP and (**C**) MjNhaP1 templates. Yellow dashes indicate possible interactions, based on the distance between residues.

**Figure 6 plants-12-02778-f006:**
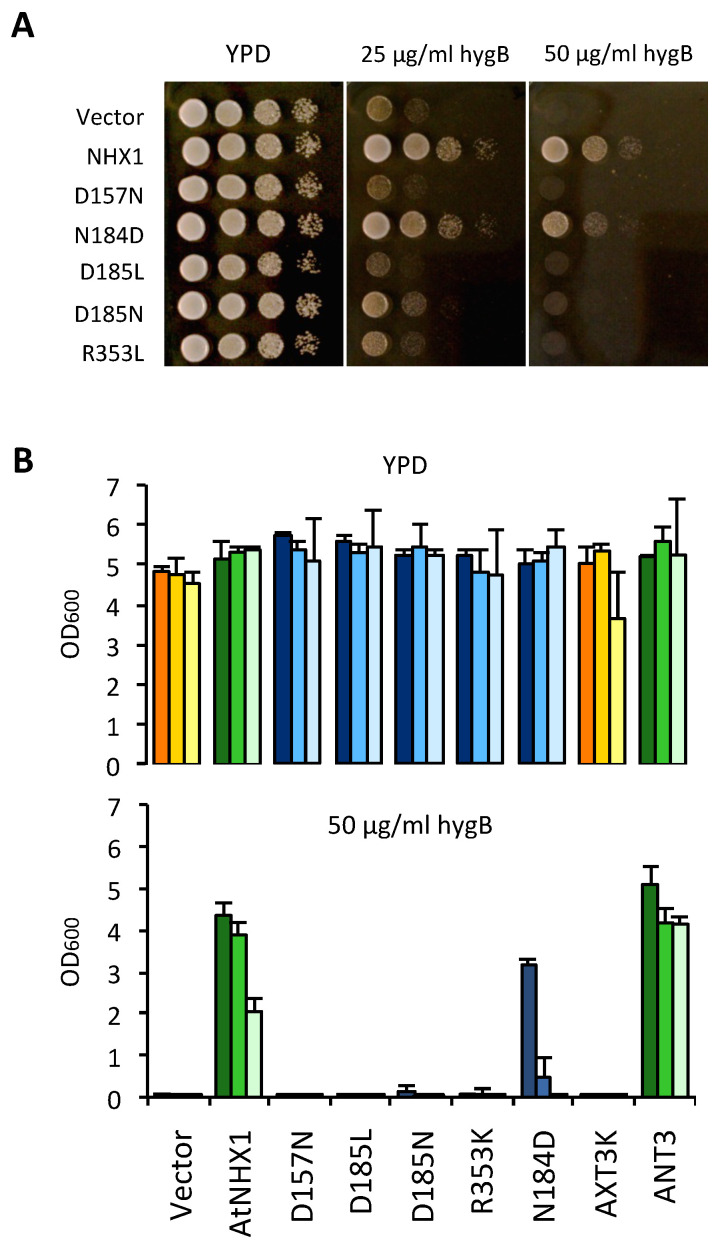
Complementation of the yeast AXT3K strain with an allelic series of AtNHX1 mutants in conserved residues. The cDNAs of AtNHX1 and the indicated mutant alleles of conserved residues were cloned into the yeast expression vector pDR195 and transformed into strain AXT3K (Δ*ena1-4* Δ*nha1* Δ*nhx1*). Overnight cultures were normalized in water to OD_600_ = 0.5. Aliquots (5 μL) from normalized cultures and 10-fold serial dilutions were spotted onto YPD solid medium plates (**A**) or used to inoculate 200 μL of YPD liquid medium in 96-well plates (**B**) with the indicated concentrations of hygromycin B. Pictures of plates were taken after 2–3 days at 30 °C. Liquid cultures were grown at 30 °C overnight before measuring the optical density. In (**B**), strain ANT3 of genotype (Δ*ena1-4* Δ*nha1 NHX1*) was used as a control expressing the endogenous ScNHX1 protein. The tree bars for each sample represent the 10-fold serial dilutions of the seed culture, from dark (10^−1^) to pale (10^−3^). The color code represents sample groups; orange are controls not expressing any NHX1 protein, green color are cells expressing AtNHX1 or ScNHX1, and the blue columns indicate AtNHX1 mutant proteins. Plots present the mean and SD of three technical replicates.

**Figure 7 plants-12-02778-f007:**
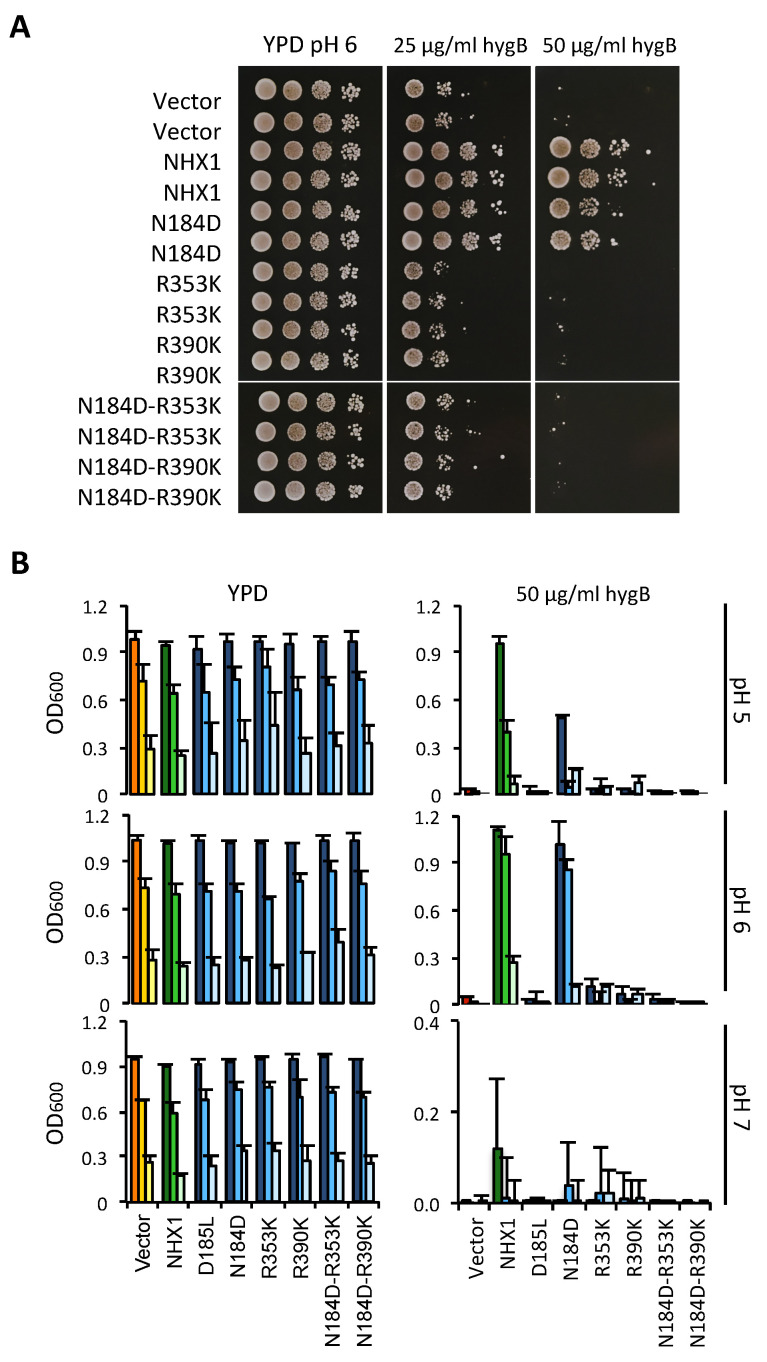
Functional assay of AtNHX1 mutant alleles in amino acids at the ion coordination pocket. The cDNAs of wild-type AtNHX1 and the indicated mutant alleles of the conserved residues were subcloned into the yeast expression vector pDR195 and transformed into the AXT3K (Δ*ena1-4* Δ*nha1* Δ*nhx1*). Overnight cultures were normalized in water to OD600 = 0.5. Aliquots (5 μL) from normalized cultures and 10-fold serial dilutions were spotted onto YPD solid medium plates (**A**) or used to inoculate 200 μL of YPD liquid medium in 96-well plates with the indicated concentrations of hygromycin B (**B**). To determine whether the mutated amino acids played a role in pH sensing, the experiment in liquid media was performed at different pHs (buffering with 10 mM MES). Plates were incubated for 2–3 days at 30 °C and pictured. The liquid cultures were grown at 30 °C overnight before measuring the OD. In (**B**), the tree bars for each sample represent the 10-fold serial dilutions of the seed culture, from dark (10^−1^) to pale (10^−3^). The color code represents sample groups; orange are controls not expressing any NHX1 protein, green color are cells expressing the wild-type AtNHX1, and the blue columns indicate AtNHX1 mutant proteins. Plots present the mean and SD of at least three technical replicates.

**Figure 8 plants-12-02778-f008:**
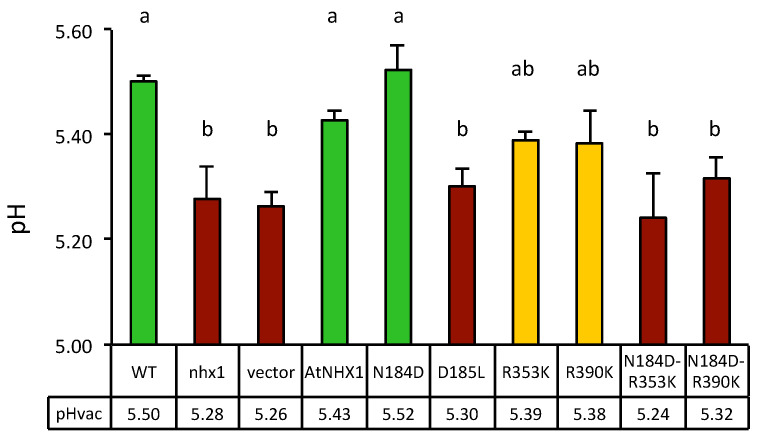
Vacuolar pH measured by BCECF-AM fluorescence in yeast expressing the AtNHX1 mutants. Vacuolar pH of the wild-type (WT) and mutant *nhx1* strains, and of *nhx1* cells expressing the plant exchanger AtNHX1 and the indicated mutant alleles, or transformed with an empty vector as negative control. Three independent transformants with 1–2 technical replicas (*n* = 5) were used for each determination. Shown are the means and SE. Lower row provides the actual mean values of pHv for each genotype. Different letters indicate statistically significant differences by Fisher LSD test (*p* < 0.01). The color code illustrates the three statistical groups (green, wild-type values; magenta, inactive mutant values; yellow, intermediate values). The experiment was repeated twice with similar results.

## Data Availability

Raw data and PDB files with protein structures are available upon reasonable request.

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
