# Peer review of "Structure-Guided Identification of Critical Residues in the Vacuolar Cation/Proton Antiporter NHX1 from Arabidopsis thaliana"

_plants, 2023, doi:10.3390/plants12152778_

Round 1
Reviewer 1 Report
Firstly, I have to say that this manuscript is not in the style of a research paper.
The contents that should be divided into methods, results, and discussion are mixed up and described in “Results” section. In the results section, you should describe your experimental results in a straightforward manner, and basically exclude any content that would require a citation. Of course, if you have some minor observations that don't need to be discussed in the Discussion section, you may include them with the cited references in Results section, but, as it stands, your interpretation accompanies every result with many citations, making it look like a "Results and Discussion". Consequently, the manuscript is very long for its content and has many duplications. Also, it is inappropriate to describe the modeling method in detail in the results section. If the modeling method itself is the subject of research, the results of modeling by multiple methods including classical methods and the newest AlphaFold2 method should be compared, and the suitability should be discussed using the data from the crystal structure analysis as the correct answer. In this respect, in this study, the modeling method cannot be the result (as you don’t have crystal structure data of AtNHX1. In silico validation does not go beyond modeling), but only a method, and should be described in a straightforward manner in the Methods section.
Secondly, and this relates to the comment immediately above, I don't think it is necessary to go into the minutiae of the modeling methodology. When I read what is written in section 2.1. for example, it is all the usual stuff for modeling, such as using AtNHX1 as bait, and the condition of the template is that the template has X-ray analysis data, and so on. It feels like an experimental lecture for undergraduate students, rather than the “Methods” I usually read in research papers. In this connection, I would like to point out that the previous studies by Yamaguchi et al (2003) and by Sato and Sakaguchi (2005), which you cited as inconclusive findings, were performed before the publication of X-ray analysis data of PaNhaP, MjNhaP, TtNapA after 2011. After all, aren't you saying that you have followed (and maybe thought out some improvement of) the standard procedure using the latest PaNhaP, MjNhaP, and TtNapA X-ray analysis data? To me, this is a clearer and more convincing motive.
I strongly recommend reconstructing the manuscript, taking care to avoid repetition as much as possible.
Other points;
The figure captions for Fig. 4 and Fig. 5 do not match each figure. Apparently, Fig. 4 doesn’t have (H), (I), (J) while you mention (H-J) in the caption.
I can not understand why the graphs in Fig.7 and Fig. 8 have 3 bars each. Write down what the dark blue, blue, and light blue indicate, respectively.
Author Response
Q1. I strongly recommend reconstructing the manuscript, taking care to avoid repetition as much as possible.
R1. Thank you for the constructive comments.
We agree with the Reviewer that the manuscript could be shortened and reorganized, which we have done for the revised version (details below). However, we also argue that what should be included or not in Results is not defined by a strict the journal policy. We favored a writing style in which we briefly explained the rationale behind each experiment, followed by the presentation of results and the interpretation of their meaning. This is a great guidance to the reader. We used Discussion for contrasting the new findings against the current knowledge.
The specific changes to the revised manuscript have been highlighted in yellow and they are:
- The Introduction has been trimmed, and the specific reference to previous results and limitations regarding AtNHX1 topology have been removed.
- Details of construction and validation of the model have been condensed and moved from Results to Materials and Methods. Former Figure 4 is now Supplemental Fig S3 and Supplemental Table S2 is not longer necessary.
- Former Sections 2.1 to 2.3 of Results have been reduced from 1642 words and 5 Figures to a single Section 2.1 with 825 words and 4 Figures.
- Former Section 3.2. Validation of structural models in Discussion has been deleted.
- References have been reduced from 76 to 65.
Regarding the specific controversy about the placement of the C-terminus of AtNHX1, we are fully aware that the technological resources we enjoy now were not the same back in 2003-2005. However, it is important to challenge the report of Yamaguchi et al (2003) because these authors proposed the radical idea that the C-terminus of AtNHX1 faced the vacuolar lumen and interacted with CaM15 (now CML18) from within the vacuole, and that this was a unique sensing mechanism for the regulation of pHv. This work is still highly cited despite the fact that it has been discredited by more recent evidence and dismissed by experts on the topic. Nevertheless, we have removed this issue from Introduction to mention it briefly only in the Discussion.
Q2. The figure captions for Fig. 4 and Fig. 5 do not match each figure. Apparently, Fig. 4 doesn’t have (H), (I), (J) while you mention (H-J) in the caption.
R2. Corrected.
Q3. I can not understand why the graphs in Fig.7 and Fig. 8 have 3 bars each. Write down what the dark blue, blue, and light blue indicate, respectively.
R3. Sorry! The meaning of the three bars per sample, the color code, and the data in plots are now explained.
Reviewer 2 Report
Please find the comments below:
Authors have addressed the structural basis for electrogenicity of Na/H exchangers. However, members of this protein group also have different substrate selectivity. NHX1 has low K-Na selectivity, whereas the related protein SOS1/NHX1 and the archaeal PaNhaP and MjNhaP1 proteins used as template here are Na-selective. Can author identify the structural determinants of substrate selectivity?
Minor issues:
Introduction. Pyrococcus abyssi and Thermus thermophilus are misspelled.
Fig 9. Is there a mistake in the labeling of statistical difference? Samples corresponding to nhx1 mutant, transformed with empty vector and complemented with AtNHX1 are all marked with 'C', but the difference seems significant when cells expressed AtNHX1.
Authors should explain why the nhx1 mutation renders the yeast cell sensitive to hygromycin.
Moderate editing of English language required
Author Response
Q1. Authors have addressed the structural basis for electrogenicity of Na/H exchangers. However, members of this protein group also have different substrate selectivity. NHX1 has low K-Na selectivity, whereas the related protein SOS1/NHX1 and the archaeal PaNhaP and MjNhaP1 proteins used as template here are Na-selective. Can author identify the structural determinants of substrate selectivity?
R1. Thank you for the constructive comments.
Certainly, the structural basis for substrate selectivity is an important issue in the structural biology of ion transporters, and specifically for the biotechnological applications of NHX proteins, which have been linked to salinity stress tolerance. We have briefly described the current knowledge in the Discussion (see new text in pages 15-16).
Q2. Introduction. Pyrococcus abyssi and Thermus thermophilus are misspelled.
R2. Corrected.
Q3. Fig 9. Is there a mistake in the labeling of statistical difference? Samples corresponding to nhx1 mutant, transformed with empty vector and complemented with AtNHX1 are all marked with 'C', but the difference seems significant when cells expressed AtNHX1.
R3. Thank you for pointing this out. There was no correspondence of statistical markings and the data shown. The mistake has been corrected in the revised manuscript. Moreover, we have incorporated additional determinations of pHvac to make the statistics more robust.
Q4. Authors should explain why the nhx1 mutation renders the yeast cell sensitive to hygromycin.
R4. Done. See new text in pg 7.
Reviewer 3 Report
1. Missing blank after “Abstract”, and “Cation/Proton” should not be bold type.
2. For reference like “Yamaguchi et al (2003) [6]”, there is no need for “(2003)”.
3. “has been described” or “have been described”.
4. Ambiguous expression for “Tridimensional structures have been generated for the tree Populus euphratica PeNHX3 using as template the structure of EcNhaA from E. coli and for CHX17 using the TtNapA structure of Thermus thermophilus as the template”
5. SwissModel is based on homology modeling, whereas the AlphaFold 2 was reported more powerful. Accordingly, AlphaFold 2 should be used to produce protein structure, at least using AlphaFold to double check the modeling result.
6. In page 7, for conservation model generation, it seems be some better if move it to materials and methods.
7. In Figure 7, mL should be used, as well as OD600. Same as in Figure 8.
8. In page 11, authors constructed several mutant alleles. Please indicate the rules for mutation. As far as I know, people usually use random mutation or basic amino acid to acidic amino acid or acidic amino acid to basic amino acid or acidic and basic amino acid to neutral amino acid. I didn’t see the information.
9. In the legend of Figure 8, “5μL” missing blank.
no
Author Response
Q1. Missing blank after “Abstract”, and “Cation/Proton” should not be bold type.
Q2. For reference like “Yamaguchi et al (2003) [6]”, there is no need for “(2003)”.
Q3. “has been described” or “have been described”.
R1-3. All corrected. See all changes highlighted in yellow.
Q4. Ambiguous expression for “Tridimensional structures have been generated for the tree Populus euphratica PeNHX3 using as template the structure of EcNhaA from E. coli and for CHX17 using the TtNapA structure of Thermus thermophilus as the template”
R4. Changed to 'Structures have been modeled for ...'
Q5. SwissModel is based on homology modeling, whereas the AlphaFold 2 was reported more powerful. Accordingly, AlphaFold 2 should be used to produce protein structure, at least using AlphaFold to double check the modeling result.
R5. Agreed. In fact, we stated in Section 3.1 that 'While this work was in progress, the modeled structure of AtNHX1 generated using the AlphaFold AI software was released', and we compared the SwissModel and AF structures in Supplemental Figure S5. The alignment of the pore domain rendered by both methods was excellent. The main difference was that the AF structure showed that the C- and N-terminal ends of the protein were located in different sides of the membrane. This was a direct consequence of the two complete TM segments (TM1 and TM2) modeled by the AlphaFold software, which in our model these segments are considered as two intramembranous (IM) semi-helices.
Q6. In page 7, for conservation model generation, it seems be some better if move it to materials and methods.
R6. Agreed. We have moved the pertinent text to Materials and Methods and the data to Supplementary Figures.
Q7. In Figure 7, mL should be used, as well as OD600. Same as in Figure 8.
R7. Corrected.
Q8. In page 11, authors constructed several mutant alleles. Please indicate the rules for mutation. As far as I know, people usually use random mutation or basic amino acid to acidic amino acid or acidic amino acid to basic amino acid or acidic and basic amino acid to neutral amino acid. I didn’t see the information.
R8. We followed different rationales for amino acid substitutions. As we explained the in the Results section, mutation N184D was made to convert the N184-D185 motif in a DD motif found in most electrogenic CPA2-type exchangers, whereas conservative mutations R353K and R390K were made to test whether the strict conservation of arginine residues was indispensable for activity or instead they could be replaced by another basic residue. Sorry we did not explain the rationale for the several amino acid changes to asparagine or leucine. We have added the following explanation to Materials and Methods:
For radical changes in amino acidic residues, polar residues were changed to Asn or Leu (D157N, D185L/N, R353L) based on the dissimilarity index D of Grantham, R. (1974). Mutation N184D was made to create the DD motif found in most electrogenic CPA2-type exchangers. Conservative mutations R353K and R390K were made to test whether the strict conservation of arginine residues was indispensable for activity or instead they could be replaced by another basic residue
Q9. In the legend of Figure 8, “5μL” missing blank.
R9. Corrected
Round 2
Reviewer 1 Report
The manuscript style was improved, the figure captions were corrected, to make it easier to read.
In P19, "24 and 48 days after the inoculation" must be a mistake. In addition, the inoculation period should be present also in each figure caption.
In my PDF reader, most of the illustration include some garbled characters. It may be just a problem with my computer, but just to be sure, please check before publication.
Author Response
Thank you again for the exhaustive revision and for picking up the clerical mistakes, which we have corrected. Please see changes marked in yellow in the revised document. Incubation times were already given in the figure captions.
Figures appear just fine in the Word and PDF files downloaded from the journal's site. We will check the galley carefully.

Reviewer 2 Report
Author have made all significant changes; therefore paper should be accepted in the current form.
Thanks
Minor editing of English language required
Author Response
Thank you for your great assistance to improve the manuscript.
Reviewer 3 Report
Good work for the revision.
Author Response

(The authors gave the same response as above.)
